# The NAE Pathway: Autobahn to the Nucleus for Cell Surface Receptors

**DOI:** 10.3390/cells8080915

**Published:** 2019-08-16

**Authors:** Poonam Shah, Alexandre Chaumet, Stephen J. Royle, Frederic A. Bard

**Affiliations:** 1Institute of Molecular and Cell Biology, Singapore 138673, Singapore; 2Centre for Mechanochemical Cell Biology, Warwick Medical School, Coventry CV4 7AL, UK; 3Department of Biochemistry, National University of Singapore, Singapore 119077, Singapore

**Keywords:** EGFR, nuclear associated endosomes (NAE), insulin receptor, nuclear envelope, SUN, translocon

## Abstract

Various growth factors and full-length cell surface receptors such as EGFR are translocated from the cell surface to the nucleoplasm, baffling cell biologists to the mechanisms and functions of this process. Elevated levels of nuclear EGFR correlate with poor prognosis in various cancers. In recent years, nuclear EGFR has been implicated in regulating gene transcription, cell proliferation and DNA damage repair. Different models have been proposed to explain how the receptors are transported into the nucleus. However, a clear consensus has yet to be reached. Recently, we described the nuclear envelope associated endosomes (NAE) pathway, which delivers EGFR from the cell surface to the nucleus. This pathway involves transport, docking and fusion of NAEs with the outer membrane of the nuclear envelope. EGFR is then presumed to be transported through the nuclear pore complex, extracted from membranes and solubilised. The SUN1/2 nuclear envelope proteins, Importin-beta, nuclear pore complex proteins and the Sec61 translocon have been implicated in the process. While this framework can explain the cell surface to nucleus traffic of EGFR and other cell surface receptors, it raises several questions that we consider in this review, together with implications for health and disease.

## 1. Introduction: Numerous Cell Surface Receptors Traffic to The Nucleus

Despite being comparatively understudied, the presence of cell surface receptors in the nucleus was reported as early as the 80s for the Insulin receptor (IR) and the epidermal growth factor receptor (EGFR) [1,2,3]. Conceptually, the traffic of receptors from the cell surface to the nucleus seems at odds with the paradigm of signals cascading from the cell surface to the nucleus. Typically, growth factors and ligands have been thought to influence cellular activities through signal transduction such as the Ras/Raf or PI3K/Akt cascades [4]. 

Yet, the nuclear localization concerns a great number of receptor tyrosine kinases (RTKs) with essential biological functions, such as the EGFR, its paralog ERB2, the fibroblast growth factor receptor 1 (FGFR-1), the vascular endothelial growth factor receptor 1 (VEGFR1) and the platelet derived growth factor receptor beta (PGDFR-b) [1,5,6,7]. Nuclear EGFR was shown to be full length, excluding the hypothesis of cleavage-mediated release from membranes. Furthermore, the presence of the corresponding ligand, EGF, argues for transfer from the surface, by opposition with alternative splicing of a cytosolic form [8].

The phenomenon is not restricted to RTKs; several G-protein coupled receptors (GPCR) such as the apelin receptor (APJ) and androgen receptor (α-AR) have also been reported to be localised to the nucleus [9,10]. Other cell surface proteins such as CD44 and the low-density lipoprotein receptor-related protein 1 (LRP1) have also been reported to traffic to the nucleus [11,12,13]. Other publications have reviewed the abundant literature on these cell surface receptors trafficking to the nucleus [9,14,15,16]. Despite this abundance of observations and the implied broad relevance of this phenomenon, two types of fundamental questions remain poorly understood: how these receptors traffic to the nucleus and what is their function in this organelle. In this review, after discussing briefly possible functions, we will focus mostly on the trafficking question and on EGFR as a model receptor. 

## 2. Proposed Nuclear Functions of Cell Surface Receptors and Functional Significance 

The nuclear localisation of cell surface receptors is not systematic nor constant but rather observed in specific tissues or conditions. For instance, EGFR was observed in the nucleus of highly proliferative cells such hepatocytes in the regenerative liver [3,14,17]. Various stimuli such as irradiation have been shown to stimulate the accumulation of receptors [18]. This suggests that nuclear translocation of these receptors is a regulatory mechanism and has evolved to transfer information inside the nucleus. 

Not surprisingly, one hypothesis is that these cell surface proteins directly affect gene transcription. EGFR has been proposed to act as a transcription factor for an extensive list of genes [19]. Consistently, EGFR has been shown to complex with chromatin [20]. Lin et al. were able to show that EGFR can bind to specific DNA sequences and activate gene transcription [21]. EGFR lacks a putative DNA binding domain, and therefore probably requires co-factors to activate transcription. Liang et al provided evidence for the interaction between EGFR and the transcription factor STAT5 [22]. Similarly, EGFR has also been shown to interact with STAT3 to induce the activation of inducible nitric oxide synthase (iNOS) [23], and to interact with E2F1 to regulate B-Myb expression [24]. Another RTK, IR, has been shown recently in a genome-wide analysis to bind to promoters of genes and interact with RNApol II. The interaction with DNA is mediated by the transcription factor host cell factor-1 (HCF-1) [25]. While most of the target genes suggested so far are transcribed by RNA pol II, the nuclear ErbB-2 was proposed to interact with RNA pol I and regulate the synthesis of ribosomal RNAs [26]. 

Overall, the genes regulated by the nuclear receptors have been correlated with the proposed function of these cell surface receptors. For instance, nuclear IR appears to control genes involved in metabolism and the response to insulin [25]. Nuclear VEGFR2 has been shown to amplify angiogenic responses and regulates its own transcription. This process requires phosphorylation of the receptor and stimulation by the VEGF ligand [27]. The cell proliferation promoter EGFR controls the expression of cyclins and cyclin dependent kinases [21]. Nuclear EGFR, in conjunction with STAT5, also controls the gene expression of Aurora-A, a kinase involved in cell cycle progression [22,28]. Similarly, nuclear FGFR-1 induces the expression of c-jun, part of the AP-1 complex that regulates the expression of cyclin D1.

More surprisingly, the nuclear EGFR could also regulate DNA replication rates directly by phosphorylating the proliferating cell nuclear antigen (PCNA). EGFR was shown to phosphorylate PCNA at residue Tyr-211, potentially stabilising PCNA on chromatin [29]. Consistently, a cell penetrating peptide, which competes for binding to PCNA at Tyr-211 with EGFR, has been reported to suppress tumour growth [30]. Perhaps even more surprising, nuclear EGFR interaction with DNA could also be involved in DNA repair. Indeed, nuclear EGFR can physically interact with the DNA dependent protein kinase (DNA-PK), an enzyme required for non-homologous end joining in double-strand breaks in DNA [18,31]. EGFR could also regulate DNA repair and synthesis through the phosphorylation of Histone H4 [32]. Consistent with these functions in the DNA repair, EGFR has reproducibly been shown to translocate to the nucleus after DNA damaging treatments such as cisplatin or radiation [18,31,33]. 

Overall, these proposed functions can explain why the nuclear accumulation of EGFR seems to be selected in cancer cells, especially in relapsing tumors. Increased levels of nuclear EGFR have been correlated with poor prognosis in many cancers [34,35,36]. It is associated with resistance to cancer treatments; for instance, Cetuximab and Gefitinib resistant cells have increased levels of nuclear EGFR [37,38]. Nuclear EGFR is in fact required for resistance to cisplatin [39]. Similarly, high levels of the nuclear ERBB2 receptor are correlated with lower overall survival in breast cancers [40]. Nuclear ERBB2 may be responsible for resistance to antibody treatments such as trastuzumab, the anti-ERBB2 monoclonal antibody commonly known as Herceptin [41].

This strong association with pathological conditions highlights the importance of understanding how EGFR and other RTKs traffic to the nucleus. A better knowledge of this trafficking pathway would also provide the tools to better separate the effects of the nuclear pool of EGFR and those induced by the signal transduction cascades initiated from the plasma membrane or endosomal pools of the receptor.

## 3. Four Models for Receptor Trafficking to the Nucleus

The mechanisms leading to the nuclear accumulation of receptors remain largely unknown. Clearly, this trafficking must involve at least one critical step: The extraction of the receptor from membranes. The transmembrane domain of the solubilised receptor must then be stabilised, presumably by some type of chaperone. Depending on where the extraction occurs, the solubilized receptor may also require active transport through the nuclear pore to accumulate in the nucleus. In the following paragraphs, we summarise four models that have been proposed (Figure 1). 

Models A and B nicknamed INTERNET and INFS postulate a long trafficking path, from endosomes to the Golgi, then from the Golgi to the Endoplasmic Reticulum (ER). This retrograde trafficking route is akin to what has been proposed for a variety of protein toxins such as Ricin, Pseudomonas Exotoxin A (PE), Shiga and Cholera, whose physiological targets are cytosolic [42,43,44,45,46]. These toxins then translocate from the lumen of the ER to the cytosol through ER-resident channels. 

The INTERNET (integral trafficking from the ER to the nuclear envelope transport) model relies on the continuity between ER membranes and nuclear envelope to postulate a transfer to the outer nuclear membrane (ONM). From there, the receptors would transfer to the inner nuclear membrane via the nuclear pore, then be extracted from the inner nuclear membrane (INM), then directly solubilised in the nucleoplasm. 

Model B, nicknamed the integrative nuclear FGFR-1 signalling (INFS) model was proposed to explain the nuclear accumulation of FGFR1. Similar to the classic toxin trafficking pathway, it postulates that after trafficking through the Golgi and the ER, extraction from membranes occurs at the ER with solubilisation in the cytosol, followed by transport of the solubilised receptor through the nuclear pore [47].

Model C, proposed in 2005 by the same group which later postulated the INTERNET model, suggested a direct interaction of endosomal derived EGFR containing vesicles with the NUP358 protein of the nuclear pore complex [48]. This interaction would be mediated by Importin-Beta. This model proposes a more direct trafficking route but left an unresolved question: How is the receptor extracted from the membrane at the nuclear pore complex? It is probably because of this problem that this model was apparently abandoned in favor of the INTERNET model. In addition, the proposed involvement of the COPI coat argued in favor of trafficking through the Golgi [49]. The involvement of the Golgi apparatus is also proposed for the PDGFR-b receptor trafficking to the nucleus [7]. These authors further suggested the involvement of the TATA element–modifying factor 1 (TMF-1), which has been shown to be involved in the Rab6-dependent trafficking from the Golgi to the ER [50].

Model D or the NAE model was recently proposed by some of the authors of this review [13]. Similar to model 3, it postulates a population of endosomes that is directed to the nucleus, the Nucleus (or Nuclear envelope) Associated Endosomes (NAE). By contrast with model 3 however, we propose that these endosomes fuse with the outer nuclear membrane into which the EGFR would then diffuse. The receptor would then translocate to the INM by way of the Nuclear Pore Complex (NPC). Once in the inner nuclear membrane, EGFR is extracted from the membrane and solubilise directly in the nucleoplasm. 

The most compelling argument for the INTERNET and INFS models would be the involvement of the Golgi apparatus and Golgi to ER trafficking steps. In a 2010 publication, Hung’s lab proposed that EGFR co-localize with a Golgi marker 15 min after EGF stimulation and with an ER marker after 30 min [49]. However, the colocalization was limited and was not quantified over time, raising concerns over the solidity of the conclusion. The authors also provided evidence that interfering with Arf1 using mutants could lead to a block in nuclear import [49]. Arf1 is a small GTPase known to regulate COPI coat dynamics and regulates Golgi to the ER retrograde traffic various other events at the Golgi [51]. An additional element was the reported interaction of EGFR with the γ-COPI subunit as seen by immunoprecipitation [49]. However, as both COPI and Arf1 have been implicated in regulating endosomal dynamics, it is not clear that these genetic evidences are necessarily implying a role for transport through the Golgi [52,53,54].

Similarly, various groups have used Brefeldin A, a potent inhibitor of GTP exchange factors for the Arfs small GTPAses and a drug inducing full disassembly of the Golgi [55]. Subcellular fractionation experiments in different studies have indicated that treating cells with BFA results in a decrease in the nuclear levels of EGFR and PDGFR [7,49]. By contrast, we did not find that the number of NAEs per cell is significantly affected after treatment with BFA, not did it affect EGFR traffic to the nucleus [13]. Another group reported that treatment with BFA does not lead to a reduction in CD44 translocation to the nucleus, whereas the microtubule inhibitor nocodazole did [11]. Thus, the evidence accumulated so far to support a long route passing by the Golgi and the ER appears limited at present. 

Still, it is possible that two routes exist, one short and one long passing by the Golgi. Perhaps the long route would result in Golgi-specific modifications, thus leading to different binding specificity. The INTERNET and INFS pathways have previously been reviewed [47,56]. In this review, we will focus on the NAE pathway; considering what we know so far, potential players, and the questions it raises. 

Several lines of evidence support the NAE model. The NAE vesicles were initially visualised with an exogenous fluorescently labeled PE toxin. PE binds to LRP1, a common cell surface receptor with multiple physiological functions [57,58]. In some cell lines, a significant accumulation of LRP1 in the nucleus can be observed [13]. PE-containing vesicles were found in close apposition to the nuclear envelope using 3D-SIM microscopy and electron microscopy. By live microscopy, the fluorescent PE content of the nuclear associated vesicles was decreasing at a constant rate, while neighboring vesicles displayed a constant signal. The interpretation of this observation is a slow and constant discharge of the NAE content into the nucleus. Such scenario is consistent with a limited fusion event and the formation of a neck between the NAE and the ONM. Such a neck structure could be observed by EM [13]. These NAE features strongly suggest that the NAE are indeed the vectors delivering PE to the nucleus. As EGFR colocalizes with PE in NAE, it suggests it follows the same path.

Following EGF stimulation, the cell surface EGFR reaches the nucleus within 5–10 min [8,59]. Exogenous PE reaches the nucleoplasm within a similar time-frame [13]. Similarly, the cell surface biotinylated IR reaches the nucleus 10 min upon insulin stimulation [8,59]). This time-frame is not consistent with a long retrograde trafficking route passing through the Golgi and the ER. By comparison, the anterograde traffic of neo-synthesised proteins such as VSVG from the ER to the cell surface takes one hour or more [60]. We and others have estimated toxin trafficking through the retrograde pathway to take a few hours (from four to six) [42,61]. This time-frame is consistent with the number of sorting and packaging steps that would be required for a receptor to percolate from endosomes to the Golgi and then to the ER, then to be extracted from the ER membrane (or lumen).

The final argument in favor of NAE is genetic. Genetic considerations support two independent routes for PE inside the cell: a slow one, following the retrograde pathway, leading to accumulation in the cytosol and inhibition of ribosomes and a rapid one, leading to the nucleus [13,42]. Several genes, such as Syntaxin 16 and ERGIC1 are specifically required for the slow trafficking of PE to the cytosol, STX16 at the Golgi level and ERGIC1 at the Golgi to ER step. These two genes are not required for traffic of PE to the nucleus. By contrast, the SUN1 and SUN2 nuclear envelope proteins are required for traffic to the nucleus of PE and EGFR but not for cytosolic delivery of PE [13,42]. 

## 4. Implications of NAE for Endosomal Sorting

Endocytosis is an essential mechanism in cells that allows internalisation of cargo such as extracellular ligands, lipids, cell-surface receptors and other macromolecules. Following internalisation, the first sorting step occurs at the early endosome [62]. From here, cargo is either (i) sent for degradation in late endosomes and lysosomes, or (ii) returned to the plasma membrane using fast or slow recycling routes, or (iii) trafficked to the Golgi apparatus [63,64]. With the NAE pathway, we need to consider a fourth fate: traffic to the nucleus. This suggests a relatively complex sorting scheme at the level of early endosomes and raises the question of the order of these sorting choices. Sorting into NAE appears to be an early event after internalisation, but more work will be required to understand the dynamics of sorting.

In endosomes, the precise trafficking route taken depends on the cargo; its sorting signals and post-translational modifications. In addition, for RTKs, endocytosis and sorting is highly influenced by interaction with their ligands. For instance, the AP2 adaptor becomes phosphorylated by EGFR upon stimulation by EGF [65]. 

Internalisation of EGFR is mainly via clathrin-mediated endocytosis (CME) with a contribution from a clathrin-independent pathway that varies with receptor density and ligand concentration [66,67]. EGFR destined for the nucleus has been proposed to be internalised specifically via CME [8]. However, for other receptors such as IR, the fraction that is fated for the nucleus can be internalised in caveolae as well as in clathrin-coated vesicles [59]. There is further complexity since the clathrin-mediated endocytosis of EGFR is subdivided by the clathrin adaptor that is used. There is AP2-dependent and AP2-independent CME of EGFR [68,69,70]. These different modes of internalisation could lead to targeting into different subtypes of early endosomes [63]. It seems clear that sorting into NAE occurs very early as NAEs co-localise with the early endosomal marker EEA1, a Rab5 effector protein [71,72]. Similarly to EGFR, GHR co-localises with early endosomal markers such as EEA1, Rab5 and Rab4 in NAE [73]. It is yet unknown whether an endosomal Rab-conversion or maturation step is involved between EE and NAE [74,75]. 

Upon stimulation, EGFR auto-phosphorylates, creating a binding site for the SH2 domain of GRB2 [76,77]. This adaptor protein then recruits the ubiquitination factor c-Cbl, which ubiquitinates EGFR [78]. There is some debate on whether ubiquitination is affecting the clathrin machinery [66,67].

Although the mechanism of endocytosis is complicated, it is clear that ubiquitylation of EGFR by the E3 ligase Cbl commits the internalised receptor to degradation. One mechanism proposed is that the ubiquitinated EGFR is recognised by Rabex-5, the ESCRT-0 component, hepatocyte growth-factor regulated tyrosine kinase substrate (Hrs), ESCRT-I component and tumor susceptibility gene 101 (Tsg101), this cascade of interactions leading to degradation [77]. Typically, degradation is thought to occur via lysosomal targeting [79]. However, targeting to the NAE will result in extraction from membranes and thus depletion from the EGFR pool able to recycle and sense signals at the cell surface. Thus, both lysosomal and nuclear targeting result in desensitization to EGF, but with presumably very different effects for the cell.

It remains to be clarified whether nuclear targeted EGFR is a ubiquitinated form. Indeed, EGF binding is not always strictly coupled with ubiquitination as deubiquitylation enzymes (DUBs) [80] can revert this fate. DUB activity enhances recycling of the receptor [81]. They could also make the receptor available for nuclear translocation. In some breast cancer cases, the deubiquitin enzymes Cezanne-1 is overexpressed, leading to an EGFR over stability in cells and perhaps promoting nuclear transport [82]. Indeed, in many cancer cells, nuclear EGFR accumulates significantly more than in normal cells [34,35]. The mechanism remains unknown, but theoretically, it could be due to a more efficient targeting to the nucleus or a slower rate of degradation upon reaching the nucleoplasmic space. Indeed, ubiquitination could also regulate the stability of the nuclear pool of EGFR by influencing its targeting to the proteasome. 

While it remains unclear how the modifications (phosphorylation, ubiquitination) of EGFR may affect its trafficking to the nucleus, it has been clearly documented that ligand binding tends to stimulate this trafficking. In fact, the majority of studies have focused on ligand binding to the receptor as a stimulant for the nuclear accumulation of receptors [8,59,83]. It has also been shown that certain ligands for EGFR including HB-EGF, TGF-α, and β-Cellulin, can induce its nuclear translocation, whereas others, amphiregulin and epiregulin stimulation cannot, even at higher concentrations [84]. Perhaps these differences are due to varying stability of the ligand-receptor interaction following endocytosis. Indeed, endosomal acidification tends to uncouple some ligands from the receptor. 

Some studies have also shown that stress such as UV rays and X-ray irradiation can also drive nuclear accumulation of EGFR, sometimes to an even stronger degree than ligand induced stimulation [31,85]. This effect of irradiation has been shown to require phosphorylation of the Thr654 residue, as deletion of this site blocked nuclear transport of EGFR [33]. Yet, it remains quite mysterious how these irradiative stimuli are transduced to the trafficking machinery and orienting the EGFR to the nucleus. 

Overall, it is clear that much remains to be discovered to understand how signaling integration can direct the three possible fates of the EGFR: Recycling to the cell surface, lysosomal degradation or targeting to the nucleus (Figure 2). 

## 5. Mechanisms at Play in the NAE Pathway

### 5.1. The Docking of NAE

When imaged by immunofluorescence, endosomes are clearly visible underneath the nucleus, in a relatively small volume sandwiched between the plasma membrane and the nuclear envelope. This suggests that a mechanism allows their concentration in this locale. Live microscopy revealed that cytoplasmic PE-loaded endosomes are undergoing rapid linear movement, consistent with a motility driven by motors along microtubules [86]. By contrast, the nuclear located PE endosomes were comparatively immotile, being three times slower on average than the cytoplasmic pool [13]. A small fraction of NAE were linearly transiting over the nucleus and abruptly stopping, suggesting they had transited from a microtubules-driven motility to a form of nuclear envelope docking.

The nuclear envelope is connected to cytoplasmic cytoskeleton elements such as actin polymers, intermediate filaments and microtubules [87]. A set of complexes mediate these connections and link the INM to the ONM. Inserted in the INM, the SUN (for Sad1 and UNC-84) proteins contain a conserved SUN domain localised in the perinuclear space and a nucleoplasmic domain that interacts with lamins [88]. The SUN domain interacts with the KASH domain of the Nesprin family of proteins. A trimeric SUN2 protein complex interacts with three Nesprin proteins [89]. Nesprins (for nuclear envelope with spectrin repeats) proteins have large cytoplasmic domains that can interact with cytoskeletal elements [90]. The SUN1 protein is required for the anchorage of Nesprin-1 and Nesprin-2 at the nuclear envelope [91,92].

In a reporter assay for nuclear accumulation of biotinylated PE, depletion of either SUN1 or SUN2 proteins lead to a dramatic reduction [13]. In this assay, results were less clear with Nesprins proteins, perhaps due to functional redundancy. Nonetheless, these results suggest that the SUN-Nesprin complex may mediate the docking of NAE to the nuclear envelope. It remains unknown, however, how they could connect to the NAE surface proteins.

### 5.2. Fusion of NAE With the Nuclear Envelope: A Hug-and-Kiss Process?

Under live-imaging, some NAE appear to be fusing with each other: After hovering in close proximity, two fluorescent structures and fluorescence pools merge [13]. This process is probably similar to the fusion of early endosomes [93]. Early endosomes contain multiple SNAREs, including members involved in exocytosis and fusion with late endosomes. Homotypic fusion relies on a complex formed by syntaxin 13, syntaxin 6, vti1a, and VAMP4 [94]. Other factors mediate the specificity of early endosomes fusion, namely the coiled-coil protein EEA1 and Rabenosyn-5 and the small GTPase Rab5 [95]. Other complexes such as CORVET mediate the tethering of early endosomes before fusion [96]. 

The inferred homotypic fusion is revealed by a local increase of fluorescence on a very short time frame (less than a minute). On the other hand, most docked NAE also display a loss of fluorescent material at a constant rate on a time scale of tens of minutes (up to 80 min). The NAE observed displayed very similar rates of fluorescence loss [13]. Our interpretation is that these NAE have fused with the ONM, but instead of a full merger with the envelope, a tubule or neck connects the two structures and allows a limited diffusion of fluorescent material in the nuclear envelope (ONM or lumenal space). Neck structures could be observed in electron microscopy images [13]. Such a tubular linkage structure could represent a diffusion barrier explaining the relatively slow rate of decrease of fluorescence. 

Such partial fusion events have been first proposed for synaptic vesicles and termed a kiss-and-run event [97]. Since then, the kiss and run model has been used to explain the degranulation of basophils and mast cells, between endosomes and lysosomes and phagosomes and endosomes [98,99]. It has also been proposed for trafficking in the secretory pathway and the Golgi apparatus [100]. In the case of NAE, the “run” part of the model appears to be lacking as the fused NAE have not been observed to detach and move away. Maybe the more poetic “Hug-and-Kiss” model proposed by Kurokawa et al. to describe ER to cis-Golgi transport would be more appropriate [101]. How such a stable yet incomplete fusion event is regulated is a fascinating question. 

As mentioned above, many SNAREs are present on early endosomes membranes. Similarly, SNAREs are present on ER membranes and involved in the fusion of Golgi derived transport carriers [102]. In particular, it has been shown in yeast that STX18 forms a complex with the SNAREs USe1 and Sec20 and proteins such as DSL-1, which can capture and mediate fusion with Sec22 containing, Golgi derived retrograde carriers [103]. As the complex is present in the perinuclear ER, it is tempting to speculate that it is also involved in the fusion of NAE at the nuclear envelope [104]. Alternatively, the fusion may be mediated by a related complex of the endosomes-ER contact site machinery, although these complexes have been associated with lipid exchange rather than membrane fusion [105].

### 5.3. Translocation of Receptors to the INM: A Role for the NPC

Following the NAE fusion to the ONM, the NAE soluble proteins can simply diffuse to channels in the INM. NAE transmembrane proteins, by contrast, must first reach the INM if they are to be extracted into the nucleoplasm. Resident INM proteins such as SUN and LEM (Lap1-emerin-MAN1) proteins are first synthesised in the rough ER; then transported to the INM through the NPC [106]. Several modes of transport to the INM are proposed to exist. In some cases, the transport appears to require diffusion through the peripheral channels of the NPC. These channels are relatively small compared to the central channel, ~10 nm versus 50 nm, suggesting that proteins with a large extraluminal domain may not go through [107]. Indeed, increasing the size of the cytoplasmic domain of INM resident proteins such as Lamin B receptor or Sun2 prevents their INM accumulation [108,109]. Some INM proteins, such as the LEM-domain containing yeast protein Heh2 may instead require active transport and binding to importin-type proteins [110]. In such cases, the complex appears to engage the central channel of the NPC [111].

These studies may help illuminate data obtained for the nuclear transport of RTKs. The nuclear transport of EGFR was shown to depend on binding to Importin-Beta [112,113]. Similarly, nuclear import of FGFR1 is also dependent on Importin-beta [114]. In addition, the group of Hung also showed that Erb-2 in a complex with Importin-beta can bind to the nuclear pore protein Nup358; which is part of the cytoplasmic filaments [48]. A conserved NLS sequence has been identified in the EGFR family proteins, and mutation of this NLS blocks the interaction of Erb2 with Importin-Beta and impairs its nuclear translocation [115,116]. The involvement of Ran-GTP has also been shown, as the expression of a dominant negative mutant inhibits Erb-2 nuclear translocation [48]. Ran has also been implicated in the nuclear localisation of orphan receptor protein kinase (Ror1) receptor [117].

The cytoplasmic domain of EGFR, at ~59 kD, is close to the limit of passive diffusion in the side channels of the NPC. However, an active transport mechanism could involve transient remodelling of the NPC and help reconcile current observations.

## 6. Extraction of Receptors from the INM

Upon reaching the INM, EGFR must be extracted from the membrane, solubilised and released in the nucleoplasm. Several lines of evidence suggest that the Sec 61 translocon is involved in this process.

The Sec61 translocon is the core of the translocation machine that imports nascent polypeptides into the ER [118]. In mammalian cells, it is a heterotrimeric complex composed of Sec61α, Sec61β, and Sec61γ; which together form a channel. On the cytosolic face, the translocon anchors ribosomes complexed with an mRNA. On the ER lumen face, the translocon delivers the nascent polypeptide to complexes such as the calnexin chaperon and the oligosaccharyltransferase [119,120].

While typically the machine functions in the cytosol to ER lumen direction, it has long been suggested it could also function in the reverse direction [121]. For instance, ERAD substrates such as the major histocompatibility complex class I or the yeast mating pheromone interact with Sec61 [122,123,124]. Since this function of Sec61 was proposed, other complexes have been proposed for the retrotranslocation of ERAD substrates [125,126]. Sec61 has also been associated with the translocation from lumen to cytosol of various toxins, including Cholera toxin and Pseudomonas Exotoxin [42,127]. 

Typically, the Sec61 translocon is localised in ER sheet membranes, where protein translation occurs primarily. However, it was also detected in the INM by the electron microscopy or split-GFP analysis [128,129]. Thus, the translocon is ideally placed and functionally relevant to mediate the last step of EGFR journey to the nucleoplasm. 

Whereas the whole complex is involved remains however unclear. RNAi depletion of each of the three subunits blocks the translocation of PE into the nucleus [13]. However, it was reported that only Sec61b interacts with EGFR and perhaps only is involved in the dislocation of EGFR [128]. 

If Sec61 is the dislocation machine for EGFR in the nuclear inner membrane, it is likely to function differently from the threading through a pore of unfolded nascent proteins [130]. Indeed, EGFR contains multiple sites of glycosylation and could therefore not be threaded as a single chain of amino-acids. ERAD is known to dislocate large bulky cargo such as viral particles and oligomeric complexes [126,131,132]. The nuclear translocation of both ligands and receptors such as EGFR and EGF or PE and LRP1 suggests that translocation could occur en-block; but it is unclear how Sec61 would function in these conditions. It also remains to be clarified what signals, if any, distinguish EGFR or LRP1 from resident INM proteins (Figure 3). 

## 7. Conclusions and Perspectives

In this review, we attempted to describe how cell surface receptors such as EGFR can be transported to the nucleus and the likely contribution to the process of the new type of endosomes called NAE. Many questions remain open and were not discussed extensively in this review. For instance, the four-way (recycling, nuclear, degradation, Golgi-targeted) sorting logic remains to be deciphered. In addition, the role and management of binding partners and post-translational modifications of the receptors are unknown: are the glycans on EGFR removed before membrane extraction or after? Are ligands such as EGF known to reach also the nucleus extracted together with EGFR or independently? Do all cell surface receptors follow the same nuclear route? What are the mechanisms stabilising the transmembrane domain of the solubilised receptors?

Perhaps the largest question of all concerns the functional significance of this pathway. Many studies have suggested an important role in the control of proliferation (for EGFR) or metabolic function (for the IR). It appears clearly that two fundamentally different modes of signaling exist: via cytoplasmic signaling cascades or via direct interactions of the solubilised receptors in the nucleus with DNA or transcription factors. How these two signaling modalities are coordinated will be a fascinating question to explore in the future. It will be very helpful to have a better understanding and better ways to block the nuclear targeting pathway. 

One can also wonder at the evolutionary origin of this nuclear targeting pathway. A recent study has proposed that drosophila EGFR may not act in the nucleus and therefore the nuclear functions of RTKs might have emerged relatively late in metazoan history [133]. However, nuclear translocation constitutes a relatively simple signaling cascade: the receptor interacts directly with its promoter targets. This simplicity and the relative ubiquity of the phenomenon suggests that it could represent an ancient and primitive pathway. 

It is certainly a mode of signaling that has eluded detailed characterisation so far, perhaps because it seems counter-intuitive. Yet, it is likely that much would be gained by better understanding the pathway. First, the pathway could be targeted to regulate signaling, to treat cancer for instance. More specific perhaps, it might be possible to hijack this pathway to bring biologics in the nucleus of cells, thus opening the possibility to directly control gene expression in target cells.

## Figures and Tables

**Figure 1 cells-08-00915-f001:**
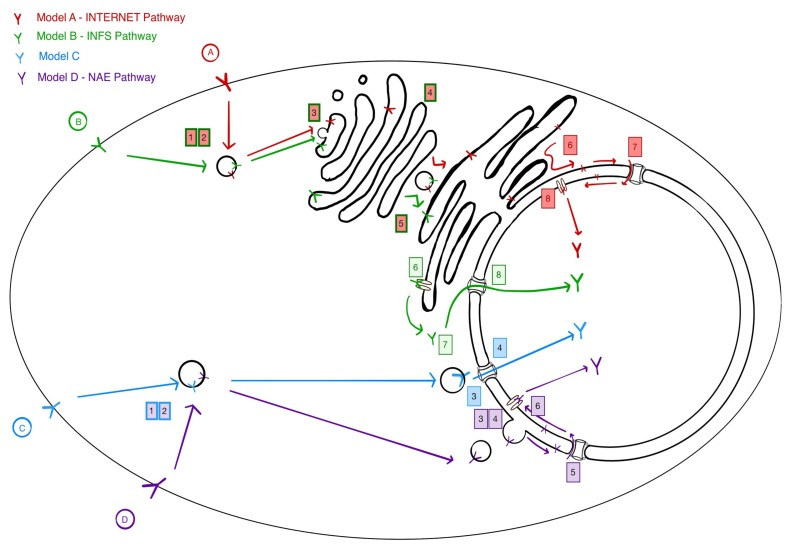
The current models to explain the nuclear translocation mechanism for cell surface receptors. The different steps required are numbered, 1: Endocytosis and 2: Sorting into the proper endosomal pathway are common to all models. Models A and B: Step 3: Targeting and fusion of endosomal carriers to Golgi membranes, 4: Retrograde Golgi traffic across the Golgi, 5: Transport from the Golgi to the ER. Model A: 6; diffusion of receptor to the outer nuclear membrane (ONM), 7: Transport across the Nuclear Pore Complex (NPC) to the inner nuclear membrane (INM), 8: Extraction to the nucleoplasm via an INM embedded channel. Model B: Step 6: Extraction to the cytosol via an ER embedded channel, 7: Binding to nuclear targeting factors, 8: Transport across the NPC. Model C: Step 3: Targeting to the nuclear envelope and interaction with the NPC, 4: Translocation and extraction across the NPC into the nucleoplasm. Model D: Step 3: Targeting and docking to the nuclear envelope, 4: Fusion of membranes and transfer of material to the outer nuclear membrane, 5: Transport across the NPC, 6: Extraction to the nucleoplasm via an INM embedded channel.

**Figure 2 cells-08-00915-f002:**
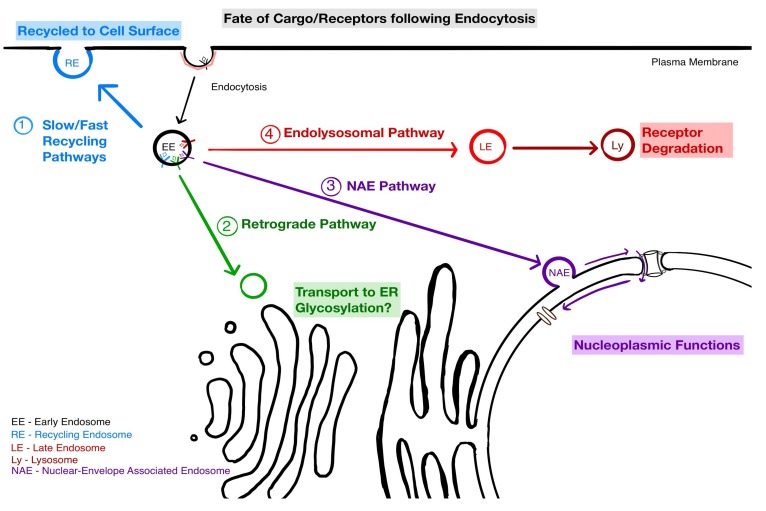
The pathways that internalised receptors can follow from the early endosomes (EE) and the fate of the receptors. (1) Recycling back to the cell surface via recycling endosomes (RE) (2); retrograde trafficking to the Golgi Apparatus (GA); (3) transport to the nucleus in nucleus associated endosomes (NAE); (4) degradation after early endosomes maturation into late endosomes (LE) and fusion with lysosomes.

**Figure 3 cells-08-00915-f003:**
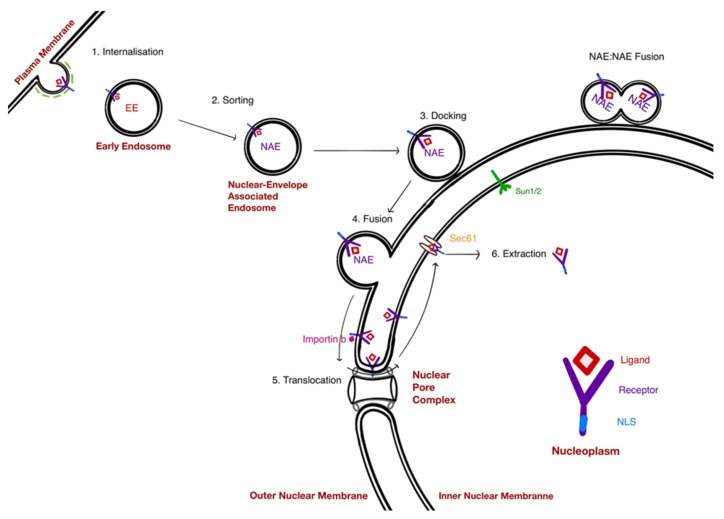
Steps in the NAE pathway. (1) Internalisation via clathrin-dependent or clathrin-independent mechanisms; (2) sorting at the early endosomal stage; (3) docking of NAE to the nuclear envelope; (4) fusion of NAE and the ONM; (5) translocation of the receptor from the ONM to the INM via the NPC; (6) extraction of the receptor from the INM membrane into the nucleoplasm.

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
