# Peer review of "The NAE Pathway: Autobahn to the Nucleus for Cell Surface Receptors"

_cells, 2019, doi:10.3390/cells8080915_

Round 1

Reviewer 1 Report

In this review, the authors address the long-standing question regarding trafficking to and function in the nucleus of cell surface receptors. After an overview of the nuclear functions of cell surface receptors, the authors turn to the trafficking of these transmembrane receptors into the nucleus, with a predominant focus on the Nuclear envelope-Associated Endosomes (NAE) pathway, which has recently been described by the authors (Chaumet et al, Nature Communications 2015). This review is a nice and timely addition and should appeal to the scientific community in this field.

Before publication, the authors should consider the focus of the review as it occasionally reads more like an opinion article based on Chaumet et al. At several points, it would be nice to see a more balanced representation of the existing data in the review.

For example, although the authors point out that several modes of transport towards the nucleus have been proposed, all modes except for the NAE pathway are largely ignored (in fact, the INFS model is consistently misspelled). The authors should compare these models and their (dis-)advantages side-by-side.

As an example, if speed is such an advantage of the NAE pathway, why is the release of PE so slow and gradual upon docking at the nuclear envelope (up to 2 hours)? And how can the continuous gradual decline in vesicle fluorescence be reconciled with a kiss-and-run or hug-and-kiss mode of contact, where one would expect a more stepwise decline in cargo load? Does this include repeated membrane neck formation and fission, or do the authors envision a stable neck combined with diffusion barriers, etc? And how would this work mechanistically? Further, can the changes in nuclear levels of cell surface receptors as observed in cancer cells, during stress or during tissue regeneration be associated with a change in preferred transport pathway?

Although the mechanistic aspects of the different stages of NAE are still unclear and under investigation the authors highlight two leads into the final stages of NAE, through regulation by Sun1, Sun2, and Sec61. As these are the only mechanistic leads to date, the authors should elaborate more on their contributions.

Here, the observation of the effects on nuclear accumulation of PE and EGFR upon depletion of SUN1 or SUN2 is interesting. Do these observations relate to the roles of SUN proteins in cytoskeletal attachments and thus docking/tethering of NAEs? Or do they reflect the role of SUN proteins in NPC association and insertion, as well as nuclear import, and as such control the nuclear levels of PE? How would this be affected by the differences in nuclear import between EGFR and PE (importin-dependent and -independent)?

Other points:

>The figures have a relatively low density of information and the authors could consider integrating them or including additional details.

>text corrections:

-page 2 bottom: ERbB2 à ErbB2

-page 3 top: highlight à highlights

-page 3 and figure 1: IFNS à INFS

-page 5 middle: propsoed à proposed

Reviewer 2 Report

Shah et al. describe briefly the possible function of the localization of the cell surface receptors inside the nucleus and summarize the different hypothetical pathways that have been proposed for the traffic of several receptor from the plasma membrane to nucleus. They focus mainly in one of these proposed models, the Nuclear Associated Endosome Model (NAE), using the EGFR as a model receptor.

The review is clear and it is well written and it would be a good way to encourage the interested reader to approach to a, certainly, obscure and controversial topic, with a plethora of unanswered question.

There are some minor points that may help to improve the manuscript:

In section 3 the authors described four different “models” to explain the traffic of the receptor to the nucleus. The general idea you get is that these four models may not be mutually exclusive, so it is not clear why the authors decided to explain in detail the NAE pathway, and it is not clear either if there would be any reason to favour any particular route over the others.

The authors information is incomplete.

Page 5 spelling mistake: “One mechanism proposed…”

            Reference missing: “DUB activity enhances recycling of the receptor (Ref)????”
